# Structure Activity Relationship and Molecular Docking of Some Quinazolines Bearing Sulfamerazine Moiety as New 3CLpro, cPLA2, sPLA2 Inhibitors

**DOI:** 10.3390/molecules28166052

**Published:** 2023-08-14

**Authors:** Mohammed Abdalla Hussein, Rita M. Borik, Mohamed S. Nafie, Heba M. Abo-Salem, Sylvia A. Boshra, Zahraa N. Mohamed

**Affiliations:** 1Biotechnology Department, Faculty of Applied Heath Science Technology, October 6 University, Giza 28125, Egypt; prof.husseinma@o6u.edu.eg; 2Chemistry Department, Faculty of Science (Female Section), Jazan University, Jazan 82621, Saudi Arabia; retag6066@hotmail.com; 3Chemistry Department (Biochemistry Program), Faculty of Science, Suez Canal University, Ismailia 41522, Egypt; mohamed_nafie@science.suez.edu.eg; 4Chemistry of Natural Compounds Department, Pharmaceutical and Drug Industries Research Institute, National Research Centre, Giza 28125, Egypt; hb_abosalem@yahoo.com; 5Department of Biochemistry, Faculty of Pharmacy, October 6 University, Giza 28125, Egypt; 6Medical Laboratory Department, Faculty of Applied Medical Sciences, October 6 University, Giza 28125, Egypt; znassar.ams@o6u.edu.eg

**Keywords:** quinazolines, sulfamerazine, 3CLpro, cPLA2, sPLA2, IL-8, TNF-α inhibitors

## Abstract

The current work was conducted to synthesize several novel anti-inflammatory quinazolines having sulfamerazine moieties as new 3CLpro, cPLA2, and sPLA2 inhibitors. The thioureido derivative **3** was formed when compound **2** was treated with sulfamerazine. Also, compound **3** was reacted with NH_2_-NH_2_ in ethanol to produce the N-aminoquinazoline derivative. Additionally, derivative **4** was reacted with 4-hydroxy-3-methoxybenzaldehyde, ethyl chloroacetate, and/or diethyl oxalate to produce quinazoline derivatives **5**, **6**, and **12**, respectively. The results of the pharmacological study indicated that the synthesized **4**–**6** and **12** derivatives showed good 3CLpro, cPLA2, and sPLA2 inhibitory activity. The IC_50_ values of the target compounds **4**–**6**, and **12** against the SARS-CoV-2 main protease were 2.012, 3.68, 1.18, and 5.47 µM, respectively, whereas those of baicalein and ivermectin were 1.72 and 42.39 µM, respectively. The IC_50_ values of the target compounds **4**–**6**, and **12** against sPLA2 were 2.84, 2.73, 1.016, and 4.45 µM, respectively, whereas those of baicalein and ivermectin were 0.89 and 109.6 µM, respectively. The IC_50_ values of the target compounds **4**–**6**, and **12** against cPLA2 were 1.44, 2.08, 0.5, and 2.39 µM, respectively, whereas those of baicalein and ivermectin were 3.88 and 138.0 µM, respectively. Also, incubation of lung cells with LPS plus derivatives **4**–**6**, and **12** caused a significant decrease in levels of sPLA2, cPLA2, IL-8, TNF-α, and NO. The inhibitory activity of the synthesized compounds was more pronounced compared to baicalein and ivermectin. In contrast to ivermectin and baicalein, bioinformatics investigations were carried out to establish the possible binding interactions between the newly synthesized compounds **2**–**6** and **12** and the active site of 3CLpro. Docking simulations were utilized to identify the binding affinity and binding mode of compounds **2**–**6** and **12** with the active sites of 3CLpro, sPLA2, and cPLA2 enzymes. Our findings demonstrated that all compounds had outstanding binding affinities, especially with the key amino acids of the target enzymes. These findings imply that compound **6** is a potential lead for the development of more effective SARS-CoV-2 Mpro inhibitors and anti-COVID-19 quinazoline derivative-based drugs. Compound **6** was shown to have more antiviral activity than baicalein and against 3CLpro. Furthermore, the IC_50_ value of ivermectin against the SARS-CoV-2 main protease was revealed to be 42.39 µM, indicating that it has low effectiveness.

## 1. Introduction

COVID-19 is the greatest pandemic outbreak of the twenty-first century and has turned into a worldwide hazard to public health [1]. 3CLpro, known as Mpro, is crucial for viral replication. One of the primary targets for the creation of anti-SARS-CoV medications is the protease 3CLpro, which generates 11 of the 16 NSPs that are produced when PLpro and 3CLpro cleave the PP chain into NSPs [2,3].

Endotoxin causes an inflammatory response in a variety of cells [4], which results in the creation of PLA2 and pro-inflammatory mediators [5]. Several reports have shown the role of sPLA2 in lung inflammation and surfactant breakdown, which may be important to protect lung tissue against COVID-19 infection [6,7]. On the other hand, the production of lysophospholipids and free fatty acids such arachidonic acid, which is a precursor to eicosanoids, is a neutral result of the hydrolysis of membrane phospholipids. Accordingly, the PLA2 superfamily has been related to lung inflammatory diseases [8] and is thought to be an indicator of inflammation [4]. These enzymes are crucial for the start and development of the inflammatory response. They cause exocytosis in macrophages and mast cell degranulation in eosinophils [4,8].

Quinazoline derivatives have been shown to possess substantial activity as antihypertensive, anti-fibrillatory, choleretic, antiphlogistic, antimitotic, antifungal, and anticonvulsant agents [9,10,11,12]. It is interesting to note that quinazoline derivatives exhibit activity against DNA viruses such as HSV and HBV, as well as RNA viruses such as HIV, EBOV, RSV, HCV, and IAV [13,14,15,16,17,18,19]. On the other hand, sulfa drugs and their metabolites can inhibit leukotrienes and prostaglandins by blocking the cyclo-oxygenase and lipoxygenase pathways [20]. The specific enzymes that have been investigated include PLA2, COX-1, COX-2, and arachidonate 5-lipoxygenase [21,22,23,24]. Inhibitory activities against other non-arachidonic acid derivatives have also been observed, including PPAR gamma, NF-Kb, and IkappaB kinases alpha and beta [25,26,27]. Incorporating these two structural characteristics into a single molecule may produce compounds with unique biological capabilities [28]. However, several studies have reported the biological functions of certain heterocyclic compounds containing nitrogen and sulfur, namely anticancer, anti-inflammatory, antibacterial, and antitumor activities [9,10,11,12]. These findings motivated us to investigate the various pharmacological properties of newly synthesized triazoloquinazolines and triazinoquinazolines containing sulfamerazine moieties as new 3CLpro, cPLA2, and sPLA2 inhibitors. Using Autodock vina software version 1.2, the putative binding interactions of the newly synthesized compounds with 3CLpro, sPLA2, and cPLA2 enzymes were determined using docking simulations and the outcomes were compared to the drug ivermectin. In addition, the ADMET properties of all the studied compounds were calculated using free online SwissADME (http://www.swissadme.ch/) and admetSAR prediction software (https://www.computabio.com/admet-prediction-service.html).

## 2. Results

### 2.1. Chemistry 

Synthesized compounds **2**–**12** were prepared according to the proposed mechanism illustrated in Figure 1a,b. IR, ^1^H-NMR, and mass spectral results of derivatives **3**–**6** and **12** are shown in Appendix A. The IR spectrum of compound **3** showed the three typical bands at 3420, 3380, and 3150 cm^−1^ for 3NH, as well as a significant band for C=S, but the NCS band was absent. Also, according to the ^1^H-NMR spectral data of compound **3**, the signals of CH_3_, OCH_3_, aromatic, pyrimidine, and SO_2_NH protons were significant. In addition, the mass spectrum of compound **3** showed a molecular ion peak at 457 and a base peak at 287.09. 

However, the IR spectrum of compound **4** indicated absorption in the region of 3301, 3205, and 3097 cm^-1^ (NH, NH_2_) and 1685 cm^−1^ (C=O). Also, the ^1^H-NMR spectral data of compound 4 showed significant signals of CH_3_, NH_2_, and SO_2_NH protons. Also, the mass spectrum of compound **4** showed a molecular ion peak at 423 and a base peak at 240. 

On the other hand, the IR spectrum of **5** revealed the presence of the NH characteristic band at 3402 cm^−1^. Also, the ^1^H-NMR data of compound **5** showed the specific signals of C=NH protons.

Also, the mass spectrum of compound **5** showed a molecular ion peak at 557 and a base peak at 139.

Also, the IR spectrum of compound **6** exhibited the bands of NH and C=O groups. Also, the present results revealed significant ^1^H-NMR signals of pyrimidine, CH_3_, NH, and SO_2_NH protons. In addition, the mass spectrum of compound **6** showed a molecular ion peak at 463 and base peak at 199. 

However, the IR spectra of compound **12** showed bands for 2NH, CH-aliph., and 2C=O. Also, the ^1^H-NMR chart showed clear signals for CH_2_, CH_3_, and SO_2_NH protons. In addition, the mass spectrum of compound **12** showed a molecular ion peak at 509 and a base peak at 60.16.

### 2.2. Biological Studies

#### 2.2.1. CLpro Inhibitory Activity of Synthesized Compounds **4**–**6** and **12**


The synthesized compounds **4**–**6** and **12** were effective inhibitors of 3CLpro (Table 1 and Figure 2), and their activity was comparable to that of ivermectin. The IC_50_ values of the target compounds **4**–**6** and **12** were 2.012, 3.68, 1.18, and 5.47 µM, respectively, whereas those of baicalein and ivermectin were 1.72 and 42.39 µM, respectively. The IC_50_ values of the synthesized compounds were less than those of the reference compounds. 

#### 2.2.2. sPLA2 Inhibitory Activity of Synthesized Compounds **4**–**6**, and **12**

The synthesized compounds **4**–**6** and **12** were effective inhibitors of sPLA2 (Table 2 and Figure 3), and this activity was comparable to that of ivermectin. The IC_50_ values of the target compounds **4**–**6** and **12** were 2.84, 2.73, 1.016, and 4.45 µM, respectively, whereas the IC_50_ values of baicalein and ivermectin were 0.89 and 109.6 µM, respectively. 

#### 2.2.3. cPLA2 Inhibitory Activity of Synthesized Compounds **4**–**6** and **12**

The synthesized compounds **4**–**6** and **12** were effective inhibitors of cPLA2 (Table 2 and Figure 4), and this activity was comparable to that of ivermectin. The IC_50_ values of the target compounds **4**–**6**, and **12** were 1.44, 2.08, 0.5, and 2.39 µM, respectively, whereas those of baicalein and ivermectin were 3.88 and 138.0 µM, respectively. The IC_50_ values of the synthesized compounds were lower than those of the reference compounds.

#### 2.2.4. Effects of Compounds **4**–**6,** and **12** as well as Baicalein and Ivermectin on sPLA2, cPLA2, IL-8, TNF-α, and NO in Isolated Lung Cells Treated with LPS

The levels of sPLA2, cPLA2, IL-8, TNF-α, and NO in LPS-treated cells were dramatically increased by 929.22%, 494.2%, 557.07%, 105.52%, and 285.55%, respectively, compared to normal non-treated isolated lung cells (Table 2) (*p* < 0.05). A one-hour incubation of rat isolated lung cells with 1 µg/mL LPS plus 1.44 µM compound **4** caused a significant decrease in sPLA2, cPLA2, IL-8, TNF-α, and NO levels by 55.29%, 61.75%, 55.57%, 33.36%, and 45.66%, respectively. Similarly, isolated cells incubated for one hour with 1 µg/mL LPS plus 2.84 µM compound **4** caused a statistically significant decrease in sPLA2, cPLA2, IL-8, TNF-α, and NO levels by 77.54%, 76.19%, 62.20%, 50.90%, and 63.29%, respectively. 

Moreover, exposure of cells with 1 µg/mL LPS for one hour plus the addition of 2.08 µM compound **5** caused a smaller statistically significant decrease in sPLA2, cPLA2, IL-8, TNF-α, and NO levels by 44.33%, 55.32%, 52.26%, 31.26%, and 33.29%, respectively, compared to the LPS-exposed cells (*p* < 0.05). However, incubation of LPS-exposed cells for one hour with 3.68 µM compound **5** caused a statistically significant decrease in sPLA2, cPLA2, IL-8, TNF-α, and NO levels by 73.63%, 74.48%, 58.19%, 46.44%, and 57.18%, respectively.

Also, incubation of lung cells with LPS plus 0.5 µM compound **6** caused a significant decrease in sPLA2, cPLA2, IL-8, TNF-α, and NO levels by 66.47%, 70.54%, 68.82%, 40.59%, and 64.03%, respectively, compared to the LPS-exposed cells (*p* < 0.05). Also, incubation of lung cells with LPS plus 1.18 µM compound **6** caused a significant decrease in sPLA2, cPLA2, IL-8, TNF-α, and NO levels by 85.88%, 80.94%, 79.62%, 56.72%, and 72.27%, respectively.

Additionally, incubation of LPS-exposed cells with 2.39 µM compound **12** caused a significant decrease in sPLA2, cPLA2, IL-8, TNF-α, and NO levels by 34.22%, 52.15%, 33.10%, 23.59%, and 26.57%, respectively, compared to the LPS-treated cells (*p* < 0.05). Also, LPS-exposed cells incubated with 5.47 µM compound **12** caused a statistically significant decrease in sPLA2, cPLA2, IL-8, TNF-α, and NO levels by 68.66%, 74.37%, 44.08%, 43.75%, and 54.89%, respectively. Moreover, treatment of LPS-exposed cells with 0.89 µM baicalein caused a marked depletion in sPLA2, cPLA2, IL-8, TNF-α, and NO levels by 58.11%, 67.33%, 64.11%, 34.76%, and 53.33%, respectively, compared to the LPS-treated cells (*p* < 0.05). Also, incubation of LPS-exposed cells with 3.88 µM baicalein revealed a significant inhibition of sPLA2, cPLA2, IL-8, TNF-α, and NO levels by 80.88%, 78.13%, 75.26%, 56.01%, and 70.07%, respectively.

However, incubation of LPS-exposed cells with 131.01 µM ivermectin caused a significant decrease in sPLA2, cPLA2, IL-8, TNF-α, and NO levels by 18.09%, 58.39%, 34.32%, 11.23%, and 9.59%, respectively, compared to the LPS-treated cells (*p* < 0.05). In addition, LPS-exposed cells incubated with 149.39 µM ivermectin caused a statistically significant decrease in sPLA2, cPLA2, IL-8, TNF-α, and NO levels by 51.41%, 64.41%, 45.99%, 27.13%, and 26.02%, respectively.

#### 2.2.5. In Silico Studies

In terms of the binding activity of the tested compounds towards SARS-CoV-2 main protease activity, they exhibited good binding energies (in the range of −12.92 to −23.28 Kcal/mol) compared to baicalein (binding energy of −15.07 Kcal/mol) Table 3. Compounds **2**–**6** and **12** exhibited an interaction binding mode by forming an H-bond with Gly 143 (backbone) as the H-bond acceptor, and some compounds exhibited additional pi–pi interactions with His 41. Additionally, they exhibited good binding energies (in the range of −11.43 to −23.57 Kcal/mol) compared to baicalein (binding energy of −18.24 Kcal/mol) towards phospholipase A2 (sPLA2) activity; all compounds exhibited an interaction binding mode by forming an H-bond with His 47. Moreover, they exhibited good binding energies (in the range of −7.22 to −17.82 Kcal/mol) compared to baicalein (binding energy of −15.54 Kcal/mol) towards cytosolic phospholipase A2 (cPLA2) activity; all compounds exhibited an interaction binding mode by forming a pi–pi interaction dipole with Tyr 96. Figure 5A–C shows the binding disposition of compound 6 towards 3Clpro, sPLA2, and cPLA2 proteins, respectively. It was deposited inside their active sites and formed good binding interactions with the key amino acids of Gly 143, His 47, and Tyr 96, with acceptable bond lengths shorter than 2.5 Å. In comparison, baicalein is shown as a comparative drug in Figure 6, as it shares some structural moiety similarities with the tested compounds as quinazoline-based derivatives. The “4H-chromen-4-one” in baicalein represents the pharmacophoric equivalent to “quinazoline” in the tested compounds through ring variation. Baicalein was used as a comparative standard drug; it showed a good interaction profile with Gly 143 inside the main protease protein, it formed two H-bonds with His 47 inside the sPLA2 protein, but it did not form any interactions inside the cPLA2 protein. In comparison, although ivermectin had good binding energies towards the tested proteins, it did not form any interactions with the key amino acids. Accordingly, the investigated compounds exhibited good binding affinities towards 3Clpro, sPLA2, and cPLA2 proteins, and these results agreed with the experimental results.

##### Drug Likeliness

The drug-likeness properties of our hit compounds were studied using free online Swiss ADME software (https://www.computabio.com/admet-prediction-service.html) and predicted according to Lipinski’s rule of five with some features that increase drug likeness. Compounds **2**–**4** and **6** were in line with Lipinski’s rule of five (Ro5) and were seen to have promising oral bioavailability for the future [29,30,31]. They were all had good permeability and absorption, with topological polar surface area (TPSA) values for drug absorption through the intestine as low as 140 [32,33]. As seen in Table 4, the compounds had 0–3 H-bond donors and 3–9 H-bond acceptors. In addition, all of the compounds tested were well tolerated by cell membranes (log P was between 1.18 and 2.88). Hence, compounds **2**–**4** and **6** were found to have promising oral bioavailability (Table 4). 

##### ADMET Prediction

Table 5 illustrates the relative ADMET profiles of the screened compounds. It was found that all of the studied compounds could penetrate the blood–brain barrier except compounds **5** and **12**. Also, all compounds had good membrane permeability (Caco-2), human intestinal absorption, and were shown to be non-inhibitors of cytochrome, one of the key enzymes involved in drug metabolism. 

It is important to mention that the solubility of compounds can be categorized according to the value of Log S; compounds are insoluble if Log S > −10, poorly soluble if Log S = −10 to −6, moderately soluble if Log S > −6 and ˂−4, soluble if Log S = −4 to −2, very soluble if Log S = −2 to –0, and highly soluble if Log S > 0. Our compounds revealed Log S values between −2.8928 and −3.9130, so all of the studied compounds were soluble. Furthermore, all of the studied compounds were AMES negative, non-carcinogenic, and had rat oral LD_50_ values between 2.4056 and 2.0649 mol/kg (Table 5).

## 3. Discussion

### 3.1. Chemistry

This study investigated the potential utility of methyl-2-isothiocyanato benzoic acid (compound **2**) to react with sulfamerazine in DMF to give derivative **3** in high yield (Figure 1a,b). The chemical structure of compound **3** was determined using elemental and spectral data. The mass spectrum of compound **3** showed a molecular ion peak *m*/*z* at 457 (M^+^, 1.70%), with a base peak *m*/*z* at 131 (100%). Through its reaction with hydrazine hydrate, compound **3** enabled the successful one-step synthesis of many heterocyclic rings. These compounds were created by removing H_2_S (one molecule), which was identified using lead acetate test paper, and then cyclizing it intramolecularly to produce the appropriate N-amino derivative, compound **4**. The mass spectrum of compound **4** revealed a molecular ion peak *m*/*z* at 423 (M^+^, 3.45%), with a base peak *m*/*z* at 240 (100%). Also, NH_2_ in position 3 of the pyrimidine ring was confirmed by reacting derivative **4** with 4-hydroxy-3-methoxybenzaldehyde in acetic acid containing fused sodium acetate. The structures of **5** and **6** derivatives were discovered using elemental analysis and spectral data. The mass spectrum of compound **5** exhibited a molecular ion peak *m*/*z* at 557 (M^+^, 3.5%), with a base peak *m*/*z* at 139 (100%). The IR spectra of these compounds revealed the lack of NH_2_ bonds and the existence of bonds for NH, CH aromatic, and SO_2_ groups. Refluxing compound **4** and ethyl chloroacetate in the presence of sodium methoxide solution gave compound **6**, not the isomeric structure of compound **7** (Figure 2). According to our results, based on the formation of sodium salt on the less basic NH and the elimination of sodium chloride followed by cyclization, the chemical structure of compound **6** was suggested rather than the structure of compound **7** [11,12]. Its mass spectrum showed a molecular ion peak *m*/*z* at 463 (M^+^, 2.1%), with a base peak at 199.9 (100%). Additionally, the C=O band appeared at low frequency (1686 cm^−1^) in the IR chart expected for the structure of compound **6**. Also, the methylene protons that appeared at 4.3 ppm in the ^1^H-NMR chart were more evidence supporting our hypothesis. When compound **4** was combined with diethyl oxalate, triazoloquinazoline derivative **12** was formed. These results agreed with the previously reported method [12]. Also, the mass spectrum of compound **12** revealed a molecular ion peak *m*/*z* at 509.67 (M^+^, 2.3%), with a base peak at 60.18 (100%).

### 3.2. Biological Studies 

SARS-CoV and MERS-CoV caused SARS epidemics in 2003 and 2012, respectively [34]. The SARS-CoV-2 virion is made up of four key structural proteins: S, N, M, and E. The S protein is split into two functional parts, called S1 and S2. S1 facilitates viral infection by connecting to the host cell’s ACE2 [35]. The N protein is also immunogenic, mostly forming nucleocapsids to protect viral RNA [36].

The host cell’s translation and processing of viral DNA yield these vital structural proteins [37]. Therefore, blocking the production of crucial SARS-CoV-2 proteins might stop the virus from replicating [38]. Key enzymes in the peptide chain processing reaction, such as 3CLpro, have attracted attention as non-structural proteins that might enable the creation of coronavirus-specific medications [39,40]. However, following stimulation with inflammatory agents such endotoxins and TNF-α to create free fatty acids and lysophospholipids by hydrolyzing glycerophospholipids, levels of sPLA2 and cPLA2 were substantially elevated in the BAL fluid and plasma of ARDS patients [41]. 

The present study was designed to synthesize quinazoline sulfonamide derivatives as promising new 3CLpro, cPLA2, and cPLA2 inhibitors. Our results showed that compounds **4**, **6**, and **12** were effective inhibitors against 3CLpro with similar effectiveness as baicalein. Our study suggests that the presence of NCS, NH, NH_2_, SO_2_, C=S, C=O, and CH_3_O in compounds **4**–**6** and **12** produces phenolate ions with a negative charge that interact electrostatically or form hydrogen or ionic bonds with the positively charged amino groups of the viral capsid (envelope), thus preventing the virus particles from attaching to the host cells.

Based on the structural features of compound **4**, the presence of the C=O and amino groups of quinoline significantly increased the activity due to lipophilic interactions with some lipophilic amino acids of the SARS-CoV-2 main protease through His 41, His 163, and Glu 166. Also, in the sulfamerazine moiety, the nitrogen atom of the pyrimidine ring formed an H-bond with Gly 143. However, in compound **5**, the hydroxyl group (*p*-hydroxyl benzene ring) formed an H-bond with Gly 143, while the NH group of the sulfamerazine moiety as well the two aromatic rings of the quinazoline moiety were linked through lipophilic interactions with His 163, Arg 188, Ser 144, and Cys 144 of the SARS-CoV-2 main protease. On the other hand, the C=O group within the quinazoline moiety (compound **6**) formed an H-bond with Gly 143 of 3CLpro, which significantly increased the activity. 

In addition, in compound **12**, the C=O from the ester group, as a derivative from the triazole ring, formed an H-bond with Gly 143, and the sulfamerazine moiety formed dipole-induced dipole interactions with His 41 of the 3CLpro binding site groove. 

The results signal that subsequent effort should focus on the alteration of quinazolines containing sulfamerazine moieties by inserting some significant hydrophobic/hydrophilic groups. However, the quinazoline derivatives demonstrated strong activity, requiring more investigation into their structure–activity relationship. Compounds **4**–**6** and **12** demonstrated the highest suppression of 3CLpro RNA production and resistance to viral exoribonuclease activity, and therefore have potential as therapeutic candidates for the treatment of viral infections. 

Similar to binding to the main protease protein, the molecular features of binding to phospholipase A2 (sPLA2) by compounds **4**–**6** and **12** included an H-bond with His 47 in addition to lipophilic interactions with Gly 29, Cys 44, Gly 22, Tye 51, Lys 52, His 27, and Tyr 21. Furthermore, compounds **4**–**6** and **12** formed ion- or dipole-induced dipole interactions with Tyr 96, in addition to lipophilic interactions with Asn 95, Tyr 96, Asn 64, Asn 65, and His 62 amino acids of cytosolic phospholipase A2 (cPLA2). The hydrophobic channel that enables the substrate to reach the active site is made up of key amino acids, which are involved in the first interaction. The stability of the ligand–enzyme complex may have been influenced by sulfur contact and several Van der Waals interactions.

Furthermore, the present findings are consistent with those of El-Sayed et al. [42] and Barakat et al. [43] regarding the inhibition by certain quinazoline and pyrimidine derivatives against sPLA2 and cPLA2. Also, the present findings are consistent with those of Ibezim et al. [44] who reported the inhibitory effect of certain sulfonamide derivatives against 3CLpro, sPLA2, and cPLA2. Our results suggest that the combination of quinazoline and sulfamerazine structural features in a single molecule increased the inhibitory activity against 3CLpro, sPLA2, and cPLA2.

#### Effects of Compounds **4**–**6** and **12** as well as Baicalein and Ivermectin on sPLA2, cPLA2, IL-8, TNF-α, and NO in Isolated Lung Cells Treated with LPS

In the present study, incubation of LPS with isolated lung cells induced the ALI model and inflammation. Inflammation of lung cells is associated with the secretion of sPLA2, cPLA2, IL-8, TNF-α, and NO. Additionally, following activation with inflammatory substances such endotoxins and TNF-α, alveolar macrophages and epithelial cells express sPLA2 and cPLA2 [45,46]. The correlation between the treatment of lung cells with LPS and chronic inflammation was reported in several studies [47,48]. Our results proved that the treatment of LPS-exposed lung cells with compounds **4**–**6** and **12** as well as baicalein and ivermectin significantly inhibited the secretion of sPLA2, cPLA2, IL-8, TNF-α, and NO. Based on the docking results and structural features of compounds **4**–**6** and **12**, the presence of the pyrimidine, benzene, and triazine rings linked with C=O, -OCH_3,_ SO_2_, and NH_2_ active groups binding to the sPLA2 and cPLA2 binding site grooves led to the decrease in their activity.

Numerous investigations have demonstrated that compounds with phenolic and sulfonamide moieties, which are abundant in both synthetic and natural materials, actively participate in the anti-inflammatory and antiviral processes of SARS-CoV-2 [49,50]. The anti-3CLpro effects of these derivatives have not previously been documented and this study may be the first of its kind. Furthermore, using a mouse pneumonia model induced by LPS showed that compounds **4**–**6** and **12** as well as baicalein and ivermectin could inhibit the infiltration of inflammatory cells and the secretion of inflammatory cytokines IL-8, TNF-α, and NO. The present results were confirmed in the study by Zhu et al. [51] who showed that phenolic natural products can reduce inflammatory cell infiltration and the release of inflammatory cytokines IL-6, IL-1α, TNF-α, and IFN-γ in bleomycin-treated rats.

### 3.3. Molecular Docking (MD) Simulations

The docking results were contrasted with those of baicalein, the first reported nonpeptidic, noncovalent inhibitor of SARS-CoV-2 Mpro discovered by the Shanghai Institute of Materia Medica, Chinese Academy of Sciences [52]. An appealing target for the development of a COVID-19 drug, the main protease is a chymotrypsin-like cysteine protease that plays a significant role in facilitating the replication and transcription of the virus [53].

The molecular docking of the newly synthesized compounds **2**–**6** and **12** with 3CLpro, sPLA2, and cPLA2was performed using PyRx tools in Autodock vina (version 1.1.2) [54]. The crystal structures of the selected enzymes were retrieved from the Protein Data bBank (http://www.rscb.org./pdb) using 5RFS code for 3CLpro in association with PCM-0102739 [55], 1DCY code for sPLA2 in complex with I3N [52], and 1CJY code for cPLA2 in association with MES [56]. The number of docked postures created for each molecule at 3CLpro’s active pocket was ten, and they were then sorted according to binding energy [57]. The posture with the lowest binding energy and the smallest RMSD was chosen as the best match and complexed with the receptor for study. The top postures’ chemical interactions and binding mechanisms were visually evaluated using Chimera-UCSF [58,59].

#### ADMET Prediction

To find compounds that have the best possibility of becoming a treatment for a particular ailment, the ADME is evaluated during the drug development process [60]. The absorption, distribution, metabolism, excretion, and toxicity of a substance are all addressed by the ADMET characteristics. There are several techniques available now, both online and offline, to anticipate the behavior of medication candidates. In this study, the admetSAR prediction tool (http://lmmd.ecust.edu.cn/admetsar1) was used for calculating the pharmacodynamic properties of the newly synthesized compounds.

There have been several defeats in the hunt for therapeutic treatments to prevent ALI caused by inflammation. Because there is no particular therapy for ALI, the mortality rate remains high [61]. The production of sPLA2 and cPLA2 as a result of lung inflammation is recognized as a significant therapeutic target [62]. 

The current study revealed that **4**, **6**, and **12** derivatives may have a unique function in the treatment of ALI caused by oxidant stress. Importantly, derivatives **4**, **6**, and **12** were similarly efficacious when provided simultaneously after a one-hour incubation with LPS, showing the possibility of preventing the formation of ALI. 

## 4. Materials and Methods

Baicalein, ivermectin, dimethyl sulfoxide, and all reagents were purchased from Sigma-Aldrich (St. Louis, MO, USA).

### 4.1. Chemistry

Synthesis and characterization of target compounds **3**–**6** and **12**.

#### 4.1.1. Synthesis and Characterization of Methyl-2-Isothiocyanato Benzoic Acid

In a refluxing flask (100 mL), methyl 2-aminobenzoate (0.01 mol) was refluxed with thiophosgene (0.01 mol) in 20 mL absolute ethyl alcohol for 4–5 h. The obtained product was recrystallized from ethanol to give methyl-2-isothiocyanato benzoic acid.

#### 4.1.2. Synthesis and Characterization of Compound **3**

The target compound was synthesized according to the method of Ghorab et al. [9] with some modifications. Briefly, in a refluxing flask (100 mL), 0.01 mol sulfamerazine and 0.01 mol methyl-2-isothiocyanato benzoic acid were refluxed in 20 mL dimethylformamide for 3-4 h. The obtained solid was then recrystallized from ethanol to produce the thioureido derivative **3.** The melting point of compound **3** was revealed to be 279–281 °C, with a yield of 74%. Also, elemental analysis of compound **3** showed that the percent composition of carbon, hydrogen, and nitrogen was 52.41%, 4.35%, and 15.55%, respectively. 

Also, the calculated content of elements in compound **3** showed that the percent composition of carbon, hydrogen, and nitrogen was 52.51%, 4.16%, and 15.32%, respectively. The IR spectrum of compound **3** showed the three typical bands at 3420, 3380, and 3150 cm^−1^ for 3NH, as well as a significant bands at 1660, 1610, 1350, 1160, and 1240 cm^−1^ for C=O, C=N, SO_2_, and C=S, respectively, but the NCS band was absent. Also, according to the ^1^H-NMR chart, δ 2.23 (s) 3H, CH3; δ 3.7 (s) 3H, OCH_3_; δ 6.9–8.17 (m) 8H, Ar H + 2H-pyrimidine; δ 10.3 and 11.6 (2s) 2H, 2NH; and δ 13.14 (s) 1H, SO_2_NH. In addition, MS (*m*/*z*) for compound **3**: 457 (M^+^, 1.70%), 415.28 (2.54%), 287.09 (100%), 228.04 (15.76%), 194.02 (58.23%), 77.02 (10.7%), and 77 (94.24%).

#### 4.1.3. Synthesis and Characterization of Compound **4**

In a clean and dried refluxing flask (100 mL) containing 30 mL of n-butanol, 0.01 mol of compound **3** was refluxed with 5 mL hydrazine hydrate (95%) for 5 h. The reaction mixture was allowed to cool before being submerged in a beaker of ice-cold water. The obtained product was recrystallized from ethanol to yield compound **4**. The melting point of compound 4 was revealed to be 205–207 °C, with a yield of 77%. However, elemental analysis of compound **4** showed that the percent composition of carbon, hydrogen, and nitrogen was 53.80%, 4.20%, and 23.27%, respectively. In addition, the calculated content of elements in compound **4** showed that the percent composition of carbon, hydrogen, and nitrogen was 53.90%, 4.02%, and 23.16%, respectively. However, the IR spectrum of compound **4** indicated absorption in the region of 3301, 3205, and 3097 cm^−1^ (NH, NH_2_); 2993 and 2889 cm^−1^ (CH-aliph.); 1685 cm^−1^ (C=O); 1593 cm^−1^ (C=N); and 1407 and 1157 cm^−1^ (SO_2_). Also, the ^1^H-NMR chart showed δ 2.4 (s) 3H, CH_3_; δ 5.6 (s) 2H, NH_2_; δ 6.0–8.00 (m) 8H, Ar H + 2H-pyrimidine; δ 11.6 (s) 1H, NH; and δ 13.11 (s) 1H, SO_2_NH. On the other hand, MS (*m*/*z*) for compound **4**: 423 (M^+^, 12.45%), 380 (8.64%), 327 (2.74%), 240 (100%), 223 (36.6%), 120 (20.08%), 65 (93.3%), and 77 (94.24%).

#### 4.1.4. Synthesis and Characterization of Compound **5**

A mixture of 0.01 mol of compound **4** and 0.01 mol of 4-hydroxy-3-methoxybenzaldehyde was refluxed in 30 mL glacial acetic acid with 0.5 g anhydrous sodium acetate. Compound **5** was obtained and recrystallized from ethanol. The melting point of compound **5** was revealed to be 150–152 °C, with a yield of 69%. On the other hand, elemental analysis of compound **5** showed that the percent composition of carbon, hydrogen, and nitrogen was 56.35%, 4.20%, and 18.00%, respectively. However, the calculated content of elements in compound **5** showed that the percent composition of carbon, hydrogen, and nitrogen was 56.04%, 4.12%, and 17.59%, respectively. The IR chart of compound **5** revealed the presence of the NH characteristic band at 3402 cm^−1^ as well as typical bands at 3232 cm^−1^ (CH-arom.); 2920 and 2842 cm^−1^ (CH-aliph.); and 1685 cm^−1^ (C=O). Also, the ^1^H-NMR data showed δ 2.5 (s) 3H, CH_3_; δ 3.84 (s) 3H, CH_3_; δ 6.8–7.91 (m) 11H, Ar H + 2H-pyrimidine; δ 9.69 (s) 1H, NH; δ 11.47 (s) 1H, C=NH; and δ 12.35 (s) 1H, SO_2_NH.

#### 4.1.5. Synthesis and Characterization of Compounds **6** and **12**

A mixture of 0.01 mol of compound **4** and 0.01 mol ethyl chloroacetate and/or diethyl oxalate were refluxed in methanol for 10 h in the presence of 0.01 mol sodium methoxide. The reaction mixture was allowed to cool before being submerged in a beaker of ice-cold water. The obtained products were recrystallized from dioxane to yield compounds **6** and **12**. The melting points of compounds **6** and **12** were revealed to be 201–203 and 110–112 °C, respectively, with yields of 68% and 81%, respectively. Also, elemental analysis of compound **6** showed that the percent composition of carbon, hydrogen, and nitrogen was 54.42%, 3.50%, and 21.40%, respectively. However, the elemental analysis of compound **12** showed that the percent composition of carbon, hydrogen, and nitrogen was 54.80%, 3.40%, and 20.20%, respectively. 

Moreover, the calculated content of elements in compound **6** showed that the percent composition of carbon, hydrogen, and nitrogen was 54.42%, 3.67%, and 21.17%, respectively. Also, the IR spectra of compound **6** exhibited a band at 3324 cm^−1^ attributed to NH, as well as typical bands at 2950 cm^−1^ (CH-arom.), 2863 cm^−1^ (CH-aliph.), and 1708 and 1686 cm^−1^ (2C=O). Also, signals at 2.3, 4.4, 7.2–8.0, 8.3, and 10.54 in the ^1^H-NMR spectrum revealed the presence of [s, 3H, CH_3_-pyrimidine], [s, 2H, CH_2_], [m, 11H, Ar-H], [m, 1H, NH], and [s, 1H, SO_2_NH] protons, respectively. In addition, MS (*m*/*z*) for compound **6**: 463 (M^+^, 2.1%), 385.3 (1.9%), 344.2 (4.8%), 315.07 (6.1%), 271.97 (33.4%), 199.9 (100%), 172.00 (52.6%), 155.98 (42.8%), 92.0 (5.1%), and 65.13 (3.1%). 

On the other hand, the calculated content of elements in compound **12** showed that the percent composition of carbon, hydrogen, and nitrogen was 54.40%, 3.50%, and 20.30%, respectively. However, the IR spectra of compound **12** showed bands at 3471 and 3317 cm^−1^ for 2NH, as well as typical bands at 2916 cm^−1^ (CH-arom.), 2819 cm^−1^ (CH-aliph.), and 1701 and 11678 cm^−1^ (2C=O). Also, the ^1^H-NMR chart showed clear signals at 1.3, 2.5, 4.3, 6.8–7.9 and 10.54 for [t, 3H, CH_3_], [s, 3H, CH_3_-pyrimidine], [q, 2H, CH_2_], [m, 11H, Ar-H], and [s, 1H, SO_2_NH] protons, respectively. In addition, MS (*m*/*z*) for compound **12**: 509.67 (M^+^, 2.3%), 386.88 (3.1%), 337.80 (7.4%), 295.8 (6.5%),262.30 (4.2%), 200.22 (46.7%), 180.09 (3.8%), 104.95 (48.3%), 90.99 (22.5%), 77.07 (12.7%), and 60.18 (100%).

### 4.2. Biological Testing

#### 4.2.1. 3CLpro Enzyme Inhibition Assay

Baicalein and ivermectin as well as the synthesized sulfa derivatives were produced as 20 mM stock solutions and kept at −80 °C until use. For the ELISA experiment, each of the test inhibitors was diluted twice to achieve final concentrations ranging from 0 to 17.5 M; each concentration of the test compounds was analyzed in triplicate.

The 3CLpro screening test was performed in accordance with the manufacturer’s protocol (Cayman Chemical, Ann Arbor, MI, USA). Briefly, 10 µL 3CLpro enzyme was mixed with 10 µL (1 M) of assay buffer, then pre-incubated with 10 µL of (0–17.5 µM) the compounds (**4**–**6** and **12**) as well as the standard drugs for 1 h. The enzymatic reaction was initiated by the addition of 10 µL of substrate, then incubated for 16–18 h at room temperature. The absorbance and emission were recorded at 360 and 460 nm, respectively. Wells with 10 µL 3CLpro-enzyme and 10 µM of substrate in 10 µL of DMSO (1%) served as positive controls with no enzyme inhibition. Wells containing 1% DMSO and 10 µM of substrate without enzyme served as negative controls. The blank values were subtracted from all sample values.

The average fluorescence for the sample was obtained, and the inhibition percentage was computed using the calculation below:$$\mathrm{Inhibition}\;\%=\frac{Corrected\;100\%\;initial\;activity-corrected\;inhibitory\;activity}{corrected\;100\%\;initial\;activity}\times 100$$

#### 4.2.2. sPLA2 Enzyme Inhibition Assay

The sPLA2 screening assay was carried out according to the manufacturer’s protocol (Cayman Chemical). Briefly, 10 μL of sPLA2 was mixed with 10 μL of dimethyl sulfoxide. In a separate experiment, 10 µL of sPLA2 was added to 10 µL standard sPLA2 inhibitor (thioetheramide, 12.5 µM) and 200 µL substrate solution (diheptanoyl thiol). Also, 10 μL sPLA2 was mixed with 10 μL of the synthesized compounds (**4**–**6** and **12**) as well as the standard drugs (baicalein and ivermectin) dissolved in dimethyl sulfoxide at different concentrations (0–17.5 µM), then 200 µL of substrate solution was added. In a separate experiment, 10 μL of assay buffer was mixed with 10 μL of dimethyl sulfoxide and 200 µL of substrate solution. The reaction was initiated by the addition 10 µL of DTNB to all samples. The plate was carefully vortexed for 10 s, then covered with a plate cover and incubated for 15 min at 25 °C. Absorbance was recorded at 558 nm using a plate reader and the IC_50_ was determined for the synthesized compounds (**4**–**6** and **12**) as well as the standard drugs.

#### 4.2.3. cPLA2 Enzyme Inhibition Assay

cPLA2 inhibition by the synthesized compounds (**4**–**6** and **12**) as well as the standard drugs (baicalein and ivermectin) was carried out according to the manufacturer’s protocol (Cayman Chemical). Briefly, 15 μL of assay buffer was mixed with 10 μL of dimethyl sulfoxide and served as a blank sample. In a separate experiment, 10 µL cPLA2 (bee venom PLA2) was added to 5 µL assay buffer and served as a positive sample. Also, 5 µL assay buffer was mixed with 10 μL of standard cPLA2 inhibitor (Calbiochem, 1.8 nM) or the synthesized compounds (**4**–**6** and **12**) as well as the standard drugs (baicalein and ivermectin) dissolved in dimethyl sulfoxide at different concentrations (0–17.5 µM), then 200 µL of substrate solution was added. In a separate experiment, 10 μL of assay buffer was mixed with 10 μL of dimethyl sulfoxide and 200 µL of substrate solution. All samples were treated with 10 μL of arachidonoyl thiol to start the reaction. The plate was carefully vortexed for 30 s, then covered with a plate cover and incubated for 15 min at 25 °C. Absorbance was recorded at 414 nm using a plate reader. The inhibition percentage was calculated by comparison with a control experiment (absence of compounds **4**–**6** and **12** and standard drugs).

#### 4.2.4. Inhibition by Synthesized Compounds (**4**–**6** and **12**) as well as Baicalein and Ivermectin against cPLA2 and sPLA2 in LPS-Treated Mouse Lung Cells

Lung cells from healthy mice aged 2 to 3 months were isolated according to the method of Wang et al. [63] with some modifications. Briefly, ketamine (100 mg/kg) and xylazine (10 mg/kg) were used to anesthetize the mice before they were slaughtered. Adult mice were freshly slaughtered, and their lungs were promptly removed and kept in 60 mm dishes with 8 mL isolation buffer. After being submerged, the tissues were placed on ice and minced using autoclaved scissors. When switching to the next mouse, the scissors were altered. Mouse lungs were placed in empty 60 mm cell culture dishes using forceps under a clean cell culture hood. A 1 mL pipet was then used to drain the media around the lung. The lung was sliced into pieces around 100 times with scissors before being digested for 45 min at 37 °C with 8 cc of collagenase I. To make sure that all the bits of lung tissue were thoroughly removed and did not collect as tissue clumps, the culture suspensions were agitated every 15 min. 

To create a single cell suspension, the lung tissue was run through a 5 mL syringe with a 20 G cannula and clumps were triturated at least 12 times [64]. The cells were suspended in Krebs-Henseleit buffer at 5 × 10^6^ cells/mL and subsequently treated for 24 h with 1 g/mL LPS (Sigma-Aldrich, St. Louis, MI, USA). In separate experiments, the cells were pre-treated with compounds **4** (1.44 and 2.84 µM)**, 5** (2.08 and 3.68 µM), **6** (0.5 and 1.18 µM), and **12** (2.39 and 5.47 µM), as well as baicalein (0.89 and 3.88 µM) and ivermectin (131.01 and 149.39 µM) for 1 h before the addition of LPS. Untreated cells (control) and cells treated exclusively with the compounds (**4**–**6** and **12**) as well as baicalein and ivermectin were used as reference samples.

After incubation, the cell supernatants were collected, centrifuged at 800× *g* for 10 min at 4 °C, the sediment was removed, and the supernatants were aliquoted and kept at 80 °C until analysis. Adherent cells were obtained by scraping them in ice-cold PBS, followed by washing and resuspension in 1 mL PBS. Also, cPLA2 and sPLA2 levels were determined using ELISA kits according to the manufacturer’s protocol (BioSource Inc., San Diego, CA, USA; Catalog No. MBS268802, respectively). Also, the IL-8 level was estimated using an ELISA kit according to the manufacturer’s protocol (BD Bioscience, Int., San Jose, CA; Catalog No. 550999). In addition, the TNF-α and NO levels were estimated using ELISA kits according to the manufacturer’s protocol (Cayman Chemical, Ann Arbor, MI, USA; Catalog No. 500850 and 780001, respectively).

### 4.3. Molecular Docking

#### 4.3.1. Molecular Docking (MD) Simulations

Compounds **2**–**6** and **12** were docked with 3CL^pro^, sPLA2, and cPLA2 using PyRx tools in Autodock vina (version 1.1.2) [65]. From the Protein Data Bank website, the structures of the selected enzymes were acquired using 5RFS code for 3CLpro in association with PCM-0102739 [66], 1DCY code for sPLA2 in complex with I3N [67], and 1CJY code for cPLA2 in association with MES [68]. Proteins were prepared using Chimera-UCSF software (https://www.cgl.ucsf.edu/chimera/) and converted to PDBQT format by PyRx software (https://pyrx.sourceforge.io/). The baicalein and compound structures were constructed using ChemDraw ultra-10.0, saved as sdf, minimized, and converted to pdbqt files using OpenBable in PyRx software. The grid boxes were created to cover the co-crystalline ligands as follows: (i) for 3CLpro: center X = −10.79, Y = 12.82, Z = 68.49 (XYZ dimension 25 × 25 × 25 Å); (ii) for sPLA2: X = −62.10, Y = 29.13, Z = 41.44 (16.10 × 13.69 × 22.42 Å); (iii) for cPLA2 X = 86.02, Y = 32.13, Z = 59.40 (11.08 × 8.31 × 13.67 Å).

#### 4.3.2. ADMET Prediction

The drug likeliness and ADMET qualities of compounds **2**–**6** and **12** were determined utilizing the SwissADME [69] and admetSAR websites. The physicochemical characteristics calculated, including molecular HBA, HBD, MW, TPSA, RB, and LogP, were analyzed taking into account Lipinski’s rule of five.

### 4.4. Statistical Analysis

The data are provided as mean ± SD for six measurements. All of the data were analyzed using SPSS version 20 software. The one-way analysis of variance (ANOVA) test was used to assess the hypotheses. Statistical significance was defined as *p*-values less than 0.05. The IC_50_ values were calculated by non-linear regression curve fit (response vs. log concentration) using Prism 9 software. The IC_50_ values were compared using paired *t*-tests. A value of *p* < 0.05 was considered significant.

## 5. Conclusions

For the purpose of evaluating the inhibitory effects of new drugs against 3CLpro, sPLA2, and cPLA2, we synthesized a variety of quinazolines carrying sulfamerazine moieties. Derivatives **4**–**6** and **12** showed good inhibitory activity against 3CLpro, sPLA2, and cPLA2, with lower IC_50_ values. The overall results suggested that compounds **3**–**6** and **12** exhibited excellent binding affinities (ranging from −7.4 to −8.9 kcalmol^−1^) compared to ivermectin (-8.0 kcalmol^−1^) and formed multiple binding modes with key residues of 3CLpro (PDB ID: 6LU7). Compound **6** had the best binding affinity of −8.9 kcalmol^−1^ and showed an excellent binding mode by establishing five H-bonds with CYS145, ASN142, GLY143, and SER144. Significantly, compound **6** also showed more potent antiviral activity than baicalein and ivermectin (IC_50_ values of 1.18, 1.72, and 42.39 μM, respectively) against 3CLpro. On the other hand, the inhibitory activity of compound **6** against cPLA2 was more pronounced than that of baicalein and ivermectin, with IC_50_ values of 0.5, 3.88, and 138 μM, respectively.

Also, the synthesized compounds **4**–**6** and **12** significantly reduced inflammation, protecting isolated lung cells against alterations in the levels of cPLA2, sPLA2, IL-8, TNF-α, and NO, as inflammatory indicators in LPS-treated lung cells, in a dose-dependent manner. Finally, compound **6** is a potential lead for the development of more effective 3CLpro inhibitors and antiviral drugs in the future.

## Figures and Tables

**Figure 1 molecules-28-06052-f001:**
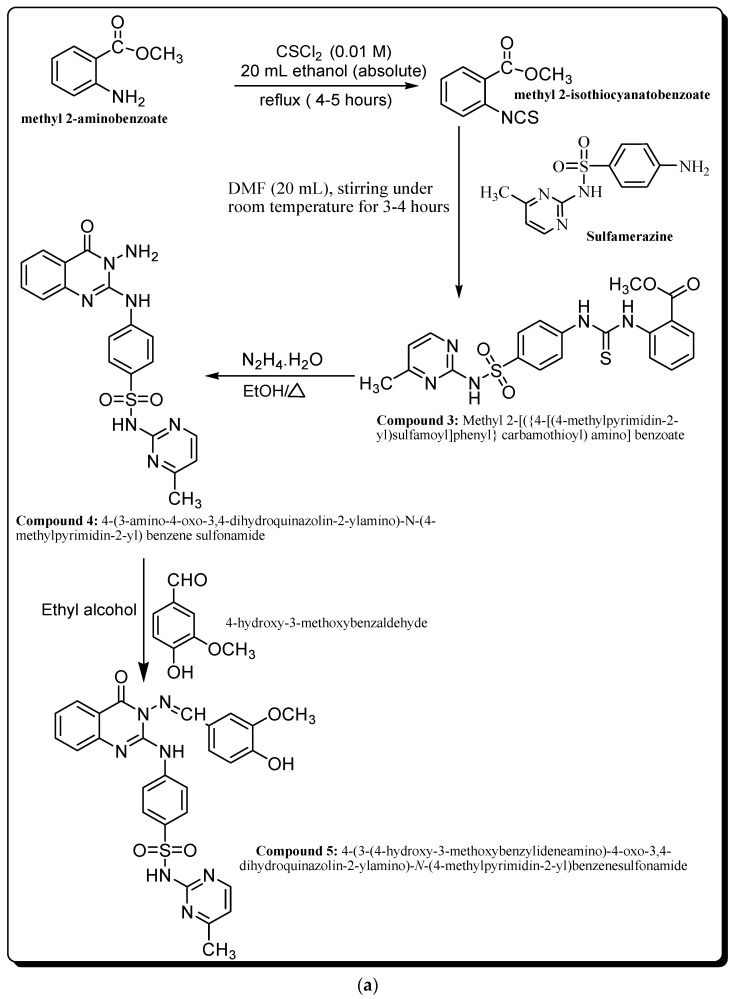
(**a**): Schematic mechanism of synthesis for the target compounds **3**–**5**. (**b**): Schematic mechanism of synthesis for the target compounds **6** and **12**.

**Figure 2 molecules-28-06052-f002:**
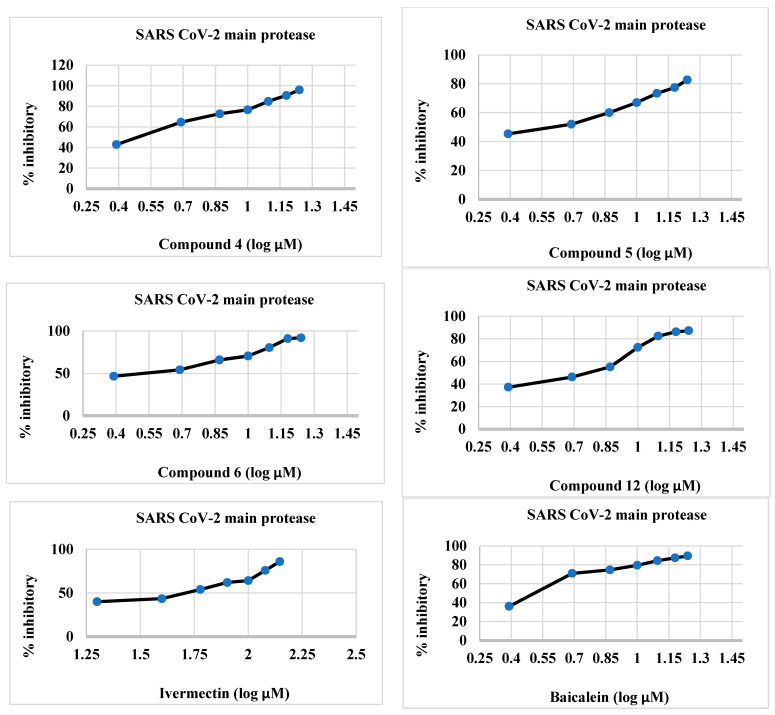
Percent inhibitory activity of synthesized compounds **4**–**6**, and **12** as well as baicalein and ivermectin against the SARS-CoV-2 main protease.

**Figure 3 molecules-28-06052-f003:**
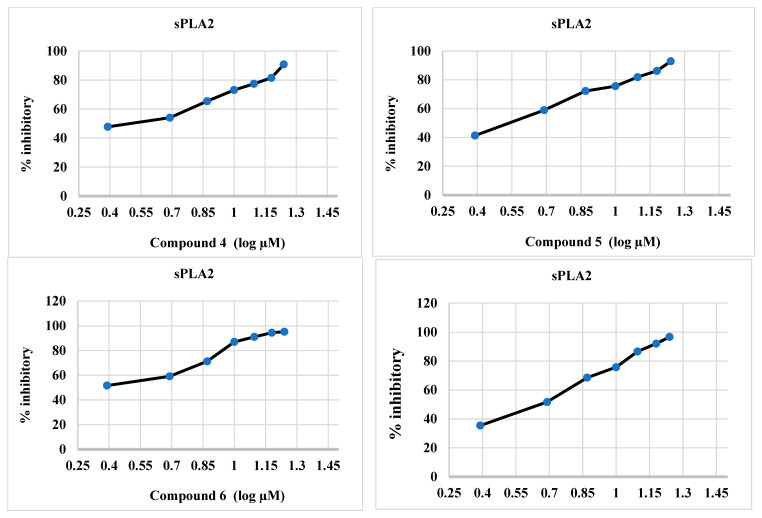
Percent inhibitory activity of synthesized compounds **4**–**6**, and **12**, baicalein, and ivermectin against sPLA2.

**Figure 4 molecules-28-06052-f004:**
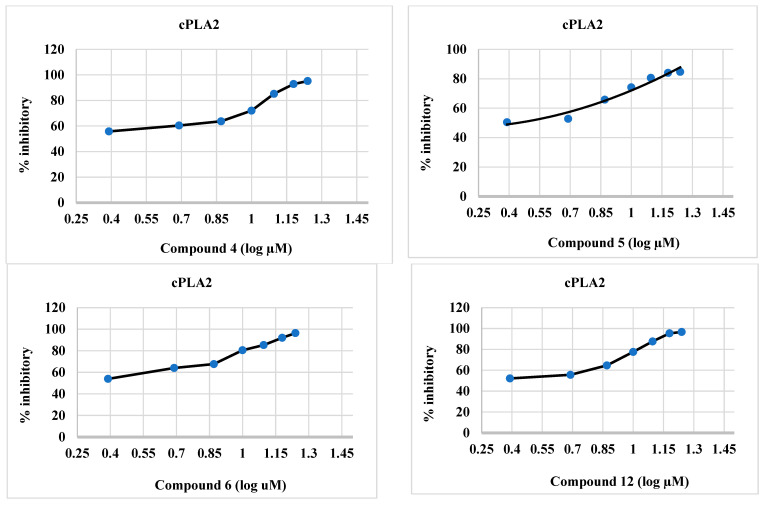
Percent inhibitory activity of synthesized compounds **4**–**6**, and **12**, baicalein, and ivermectin against cPLA2.

**Figure 5 molecules-28-06052-f005:**
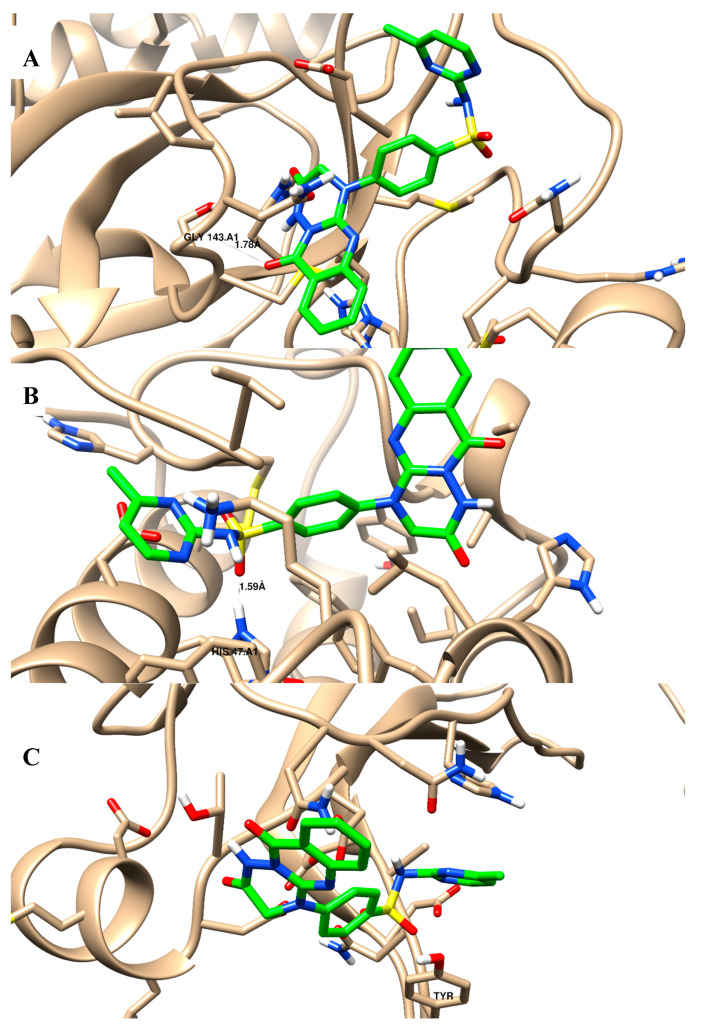
Binding disposition and ligand–receptor interactions of the docked compound **6** (green-highlighted) towards the binding sites of: (**A**) SARS-CoV-2 main protease (3Clpro) (PDB = 5RFS); (**B**) phospholipase A2 (sPLA2) (PDB = 1DCY); and (**C**) cytosolic phospholipase A2 (cPLA2) (PDB = 1CJY). Three-dimensional images were generated using Chimrea-UCSF.

**Figure 6 molecules-28-06052-f006:**
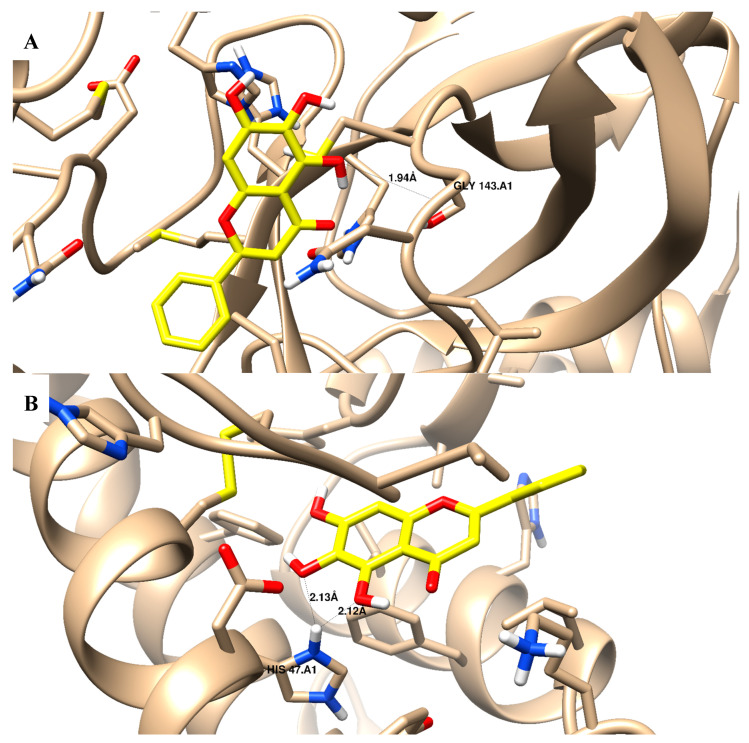
Binding disposition and ligand–receptor interactions of baicalein (yellow-highlighted) towards the binding sites of: (**A**) SARS-CoV-2 main protease (3Clpro) (PDB = 5RFS); (**B**) phospholipase A2 (sPLA2) (PDB = 1DCY). Three-dimensional images were generated using Chimrea-UCSF.

**Table 1 molecules-28-06052-t001:** IC_50_ values of synthesized compounds **4**–**6** and **12** as well as ivermectin.

Synthesized Compounds/Drugs	SARS-CoV-2 Main Protease (µM)	sPLA2 (µM)	cPLA2 (µM)
**Compound 4**	2.012 ± 0.004 ^c^	2.84 ± 0.026 ^c^	1.44 ± 0.009 ^b^
**Compound 5**	3.68 ± 2.35 ^d^	2.73 ± 0.008 ^c^	2.08 ± 0.016 ^c^
**Compound 6**	1.180 ± 0.025 ^a^	1.016 ± 0.039 ^b^	0.5 ± 0.008 ^a^
**Compound 12**	5.47 ± 0.018 ^e^	4.45 ± 0.007 ^d^	2.39 ± 0.046 ^d^
**Baicalein**	1.72 ± 0.006 ^b^	0.89 ± 0.041 ^a^	3.88 ± 0.013 ^e^
**Ivermectin**	42.39 ± 2.50 ^f^	109.6 ± 3.27 ^e^	138.0 ± 1.54 ^f^

Data are shown as mean ± standard deviation of the number of observations within each treatment. Data followed by different superscript letters along the same column are significantly different (*p* < 0.05).

**Table 2 molecules-28-06052-t002:** Effects of synthesized compounds (**4**–**6** and **12**), baicalein, and ivermectin on cPLA2, sPLA2, IL-8, TNF-α, and NO in LPS-treated isolated lung cells.

Group No.	Synthesized Compounds/Drugs	Doses	sPLA2 (pg/mL)	cPLA2 (dpm/mL)	IL-8 (ng/mL)	TNF-α (pg/mL)	NO (µmol/L)
**I**	**Negative control sample**	**0 μg/ML**	12.59 ± 0.46 ^a^	474.22 ± 25.21 ^a^	0.82 ± 0.08 ^a^	476.97 ± 32.77 ^a^	17.79 ± 2.16 ^a^
**II**	**Positive control (LPS)**	**(1 μg/mL)**	129.58 ± 6.34 ^n^	2817.86 ± 40.51 ^l^	5.74 ± 0.47 ^h^	980.27 ± 75.96 ^l^	68.59 ± 5.82 ^j^
**III**	**LPS (1 μg/mL)** **+ Compound 4**	**1.44 µM**	57.93 ± 2.92 ^i^	1077.77 ± 68.61 ^i^	2.55 ± 0.26 ^c^	654.20 ± 22.28 ^h^	37.27 ± 3.00 ^c^
		**2.84 µM**	29.10 ± 2.52 ^d^	671.05 ± 29.17 ^d^	2.17 ± 0.07 ^d^	481.36 ± 39.63 ^d^	25.18 ± 2.98 ^d^
**IV**	**LPS (1 μg/mL)** **+ Compound 5**	**2.08 µM**	72.13 ± 3.94 ^k^	1258.99 ± 35.57 ^k^	2.74 ± 0.32 ^c^	671.91 ± 23.11 ^i^	45.75 ± 2.87 ^g^
		**3.68 µM**	34.17 ± 1.73 ^e^	719.32 ± 37.15 ^e^	2.40 ±0.09 ^e^	524.98 ± 19.11 ^e^	29.37 ± 2.21 ^f^
**V**	**LPS (1 μg/mL)** **+ Compound 6**	**0.5 µM**	43.44 ± 2.74 ^g^	830.11 ± 34.00 ^f^	1.79 ± 0.14 ^d^	582.31 ± 52.25 ^e^	24.67 ± 2.70 ^e^
		**1.18 µM**	18.29 ± 2.35 ^b^	536.93 ± 23.50 ^b^	1.17 ± 0.06 ^b^	424.25 ± 36.86 ^f^	19.02 ± 1.5 ^b^
**VI**	**LPS (1 μg/mL)** **+ Compound 12**	**2.39 µM**	85.23 ± 6.51 ^l^	1348.21 ± 50.26	3.84 ± 0.29 ^g^	748.98 ± 31.22 ^j^	50.36 ± 3.96 ^h^
		**5.47 µM**	40.61 ± 4.19 ^f^	722.24 ± 30.40 ^e^	3.21 ± 0.21 ^f^	551.52 ± 24.65 ^f^	30.94 ± 3.76 ^f^
**VII**	**LPS (1 μg/mL)** **+ Baicalein**	**0.89 µM**	54.27 ± 3.46 ^h^	903.43 ± 35.71 ^g^	2.06 ± 0.23 ^g^	639.55 ± 37.84 ^g^	32.01 ± 4.81 ^f^
		**3.88 µM**	24.77 ± 2.73 ^c^	616.34 ± 28.34 ^c^	1.42 ± 0.13 ^c^	431.21 ± 32.88 ^c^	20.53 ± 1.89 ^c^
**VIII**	**LPS (1 μg/mL)** **+ Ivermectin**	**131.01 µM**	106.13 ± 9.81 ^m^	1172.50 ± 72.55 ^j^	3.77 ± 0.62 ^g^	870.22 ± 40.62 ^k^	62.01 ± 4.81 ^i^
		**149.39 µM**	62.31 ± 1.41 ^j^	1002.65 ± 90.98 ^h^	3.10 ± 0.1 ^f^	714.29 ± 28.09 ^g^	50.74 ± 3.34 ^h^

Data are shown as mean ± standard deviation of the number of observations within each treatment. Data followed by different superscript letters along the same column are significantly different (*p* < 0.05). Highly significant levels of the parameters are in the order of a < b < c < d. Data with superscript letter “a” are significantly lower than data with superscript letter “b”, while data with superscript letter “b” are lower than data with superscript letter “c” at *p* < 0.05. Data followed by the same superscript letter are not significantly different at *p* ≤ 0.05.

**Table 3 molecules-28-06052-t003:** Summary of docking results of ligand–receptor interactions and binding energies (Kcal/mol) of the docked compounds (**2**–**6**, **12**) with the key amino acids compared to baicalein *.

CPDS	SARS-CoV-2 Main Protease (3CLpro) (PDB = 5RFS) ^#^		Phospholipase A2 (sPLA2) (PDB = 1DCY)		Cytosolic Phospholipase A2 (cPLA2) (PDB = 1CJY)	
	Binding Energy (Kcal/mol)	Binding Interactions	Binding Energy (Kcal/mol)	Binding Interactions	Binding Energy (Kcal/mol)	Binding Interactions
**2**	−12.92	1 H-bond with Gly 143 pi–pi interaction with His 41	−13.19	1 H-bond with His 47	−7.22	Cation–arene interaction with Tyr 96
**3**	−19.91	1 H-bond with Gly 143	−11.43	1 H-bond with His 47	−11.17	pi–pi interaction with Tyr 96
**4**	−26.06	1 H-bond with Gly 143	−22.33	1 H-bond with His 47	−17.82	pi–pi interaction with Tyr 96
**5**	−25.83	1 H-bond with Gly 143	−23.57	1 H-bond with His 47	−13.77	pi–pi interaction with Tyr 96
**6**	−23.14	1 H-bond with Gly 143	−18.38	1 H-bond with His 47	−9.99	pi–pi interaction with Tyr 96
**12**	−23.28	1 H-bond with Gly 143 pi–pi interaction with His 41	−23.49	1 H-bond with His 47	−12.84	pi–pi interaction with Tyr 96
**Ivermectin**	−23.37	--	−6.09	--	−16.63	--
**Baicalein**	−15.07	1 H-bond with Gly 143	−18.24	2 H-bond with His 47	−15.54	--

* Docking results were visualized using Chimera-UCSF-1.15 software. ^#^ H-bond acceptor with Gly 143 (backbone).

**Table 4 molecules-28-06052-t004:** Physicochemical properties of the newly synthesized compounds **2**–**6** and **12** using the Swiss ADME online server.

Comp. No	MW g/mol	Log p	HBA	HBD	TPSA Å^2^	MR	nRB	No. Lipinski Violations
**2**	193.22	2.88	3	0	70.75	52.41	3	0
**3**	457.53	2.35	6	3	162.78	121.31	9	0
**4**	423.45	1.18	6	3	153.27	113.36	5	0
**5**	557.58	2.90	9	3	169.07	151.17	8	2
**6**	463.47	1.19	7	2	147.56	127.12	4	1
**12**	505.51	2.09	9	1	158.82	129.85	7	2

**MW**: Molecular weight; **HBA**: Hydrogen bond acceptor; **HBD**: Hydrogen bond donor; **TPSA**: Topological polar surface area; **MR**: Molar refractivity; **nRB**: Number of rotatable bonds. Drug likeness (Lipinski Pfizer filter) limits are “Yes, drug like” for MW ≤ 500, Log *p* ≤ 4.15, HBA ≤ 10, and HDD ≤ 5.

**Table 5 molecules-28-06052-t005:** Prediction of some of ADMET properties of hit compounds using the admetSAR server.

Models	2	3	4	5	6	12
**Blood–Brain Barrier**	BBB+	BBB−	BBB+	BBB−	BBB+	BBB−
**Human Intestinal Absorption**	HIA+	HIA+	HIA+	HIA+	HIA+	HIA+
**Caco-2 Permeability**	Caco-2-	Caco-2-	Caco-2-	Caco-2-	Caco-2-	Caco-2-
**Aqueous solubility (Log S)**	−2.8928	−3.2434	−3.1843	−3.3822	−3.5571	−3.9130
**P-glycoprotein Substrate**	Non-substrate	Non-substrate	Non-substrate	Non-substrate	Substrate	Non-substrate
**P-glycoprotein Inhibitor I**	Non-inhibitor	Non-inhibitor	Non-inhibitor	Non-inhibitor	Non-inhibitor	Non-inhibitor
**P-glycoprotein Inhibitor II**	Non-inhibitor	Non-inhibitor	Non-inhibitor	Inhibitor	Non-inhibitor	Non-inhibitor
**Renal Organic Cation Transporter**	Non-inhibitor	Non-inhibitor	Non-inhibitor	Non-inhibitor	Non-inhibitor	Non-inhibitor
**Subcellular localization**	Mitochondria	Mitochondria	Mitochondria	Mitochondria	Mitochondria	Mitochondria
**CYP450 2C9 Substrate**	Non-Substrate	Non-Substrate	Non-Substrate	Non-Substrate	Non-Substrate	Non-Substrate
**CYP450 2D6 Substrate**	Non-Substrate	Non-Substrate	Non-Substrate	Non-Substrate	Non-Substrate	Non-Substrate
**CYP450 3A4 Substrate**	Non-Substrate	Non-substrate	Non-substrate	Non-substrate	Substrate	Non-substrate
**CYP450 1A2 Inhibitor**	Inhibitor	Non-inhibitor	Non-inhibitor	Non-inhibitor	Non-inhibitor	Non-inhibitor
**CYP450 2D6 Inhibitor**	Non-inhibitor	Non-inhibitor	Non-inhibitor	Non-inhibitor	Non-inhibitor	Non-inhibitor
**CYP450 3A4 Inhibitor**	Non-inhibitor	Non-inhibitor	Non-inhibitor	Non-inhibitor	Non-inhibitor	Inhibitor
**CYP450 2C19 Inhibitor**	Non-inhibitor	Non-inhibitor	Non-inhibitor	Non-inhibitor	Non-inhibitor	Non-inhibitor
**Human Ether-a-go-go-Related Gene Inhibition I**	Weak inhibitor	Weak inhibitor	Weak inhibitor	Weak inhibitor	Weak inhibitor	Weak inhibitor
**Human Ether-a-go-go Related Gene Inhibition II**	Non-inhibitor	Non-inhibitor	Non-inhibitor	Non-inhibitor	Non-inhibitor	Non-inhibitor
**AMES Toxicity**	Non AMES toxic	Non AMES toxic	Non AMES toxic	Non AMES toxic	Non AMES toxic	Non AMES toxic
**Carcinogens**	Non-carcinogen	Non-carcinogen	Non-carcinogen	Non-carcinogen	Non-carcinogen	Non-carcinogen
**Honeybee toxicity**	High HBT	Low HBT	Low HBT	Low HBT	Low HBT	Low HBT
**Biodegradation**	Not readily biodegradable	Not readily biodegradable	Not readily biodegradable	Not readily biodegradable	Not readily biodegradable	Not readily biodegradable
**Acute oral toxicity**	III	III	III	III	III	III
**Carcinogenicity (three class)**	Non-required	Non-required	Non-required	Non-required	Non-required	Non-required
**Rat Acute Toxicity (LD_50_, mol/kg)**	2.2363	2.1081	2.0649	2.2374	2.2163	2.4056

## Data Availability

All data and analyses are available from the corresponding author on reasonable request.

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
