# Peer review of "Structure Activity Relationship and Molecular Docking of Some Quinazolines Bearing Sulfamerazine Moiety as New 3CLpro, cPLA2, sPLA2 Inhibitors"

_molecules, 2023, doi:10.3390/molecules28166052_

Round 1

Reviewer 1 Report (Previous Reviewer 1)

Authors synthesized several novel anti-inflammatory quinazolines having a sulfamerazine moiety as new 3CLpro, cPLA2, and sPLA2 inhibitors. The results of the pharmacological study indicated that synthesized 4-6 and 12 derivatives showed good 3CLpro, cPLA2, sPLA2 inhibitory activity. They also performed bioinformatic investigations to establish the possible binding interactions between the newly synthesized compounds 2-6 & 12 and the active site of 3CLpro. This manuscript can be accepted after major revision.

Here are the points:

-There are four values for three compounds in the sentence of Abstract Line 27 “The IC50 value of the target compounds 4-6 and 12 against sPLA2 were 2.84, 2.73, 1.016 and 4.45 µM”.

-“Significantly, compound 6 also showed more potent antiviral activity than baicalein and against 3CLpro” this sentence must be rewritten.

-“ Additionally, Because the IC50 value of  Ivermectin against SARSCoV-2 main protease was found as 42.39, which is low effective.” This sentence must be rewritten and they should add the unit.

-In Materials and methods, authors should also mention the reaction of methyl 2-aminobenzoate with thiophosgene as they put in Figure 1a.

-The NMR values are not clear in Supplementary file. Authors should rescan them.

Minor editing of English language required.

Author Response

Rebuttal Letter

Dear Editor,     

On behalf of the authors, we are delighted for considering our work for publishing in Molecules after revision and much thankful for the editor’s and reviewers’ diligent effort in helping to improve our manuscript. In the following document we clarify the inquiries raised during the reviewing cycle and provide our responses for each one raised. The reviewers’ comments were taken into consideration and a detailed clarification is addressed to each reviewers note herein and the corrections were highlighted.

No.

Reviewer 1

Responses

1

Authors synthesized several novel anti-inflammatory quinazolines having a sulfamerazine moiety as new 3CLpro, cPLA2, and sPLA2 inhibitors. The results of the pharmacological study indicated that synthesized 4-6 and 12 derivatives showed good 3CLpro, cPLA2, sPLA2 inhibitory activity. They also performed bioinformatic investigations to establish the possible binding interactions between the newly synthesized compounds 2-6 & 12 and the active site of 3CLpro. This manuscript can be accepted after major revision.

Here are the points:

-There are four values for three compounds in the sentence of Abstract Line 27 “The IC50 value of the target compounds 4-6 and 12 against sPLA2 were 2.84, 2.73, 1.016 and 4.45 µM”.

Done, compounds 4-6 and 12 rewritten to  4, 5, 6 and 12

2

-“Significantly, compound 6 also showed more potent antiviral activity than baicalein and against 3CLpro” this sentence must be rewritten.

Done , the sentence was corrected

3

-“ Additionally, Because the IC50 value of  Ivermectin against SARSCoV-2 main protease was found as 42.39, which is low effective.” This sentence must be rewritten and they should add the unit.

Done , the sentence was corrected

4

In Materials and methods, authors should also mention the reaction of methyl 2-aminobenzoate with thiophosgene as they put in Figure 1a.

Done reaction of methyl 2-aminobenzoate with thiophosgene was added in methods

5

The NMR values are not clear in Supplementary file. Authors should rescan them.

Done we will send the spectral data scan figure each compound separately

6

Minor editing of English language required.

Done, the paper language was revised carefully

Reviewer 2 Report (New Reviewer)

The presented study describes the synthesis of sulfamerazine derivatives as 3CLpro,  cPLA2, and sPLA2 inhibitors, and synthesis, biological testing and in silicon experiments were performed.

1. The choice of Baicalein (as control) for in silico studies needs more rationality as the structure of designed compounds (6, as representative compound) does not match structurally with the structure of Baicalein (or have distinctive scaffolds); therefore, direct comparison with Baicalein requires more structural information. For example, Baicalein is a flavonoid, while compound 6 has a hybrid structure and a distinctive structure than Baicalein. Therefore, comparing their binding pose and binding energies with each other seems trivial.

2. Comment specific to the author's statement, "All compounds exhibited interaction biding mode, they formed H-bond with Gly 143, some compounds exhibited additional interactions of induced-induced interactions with His 41."

(I) Please correct the typographical error "BINDING."

(ii) "H-bond with Gly 143" needs more information (was it the backbone of Gly 143 to which compounds are interacting)?

(iii) "induced-induced interactions" what are induced-induced interactions? Are authors talking here about pi-pi interaction, it seems quite confusing. Please specify such interaction influence on the molecular binding conformation of a molecule.

(3) the author made typographical errors in various places, such as it must be Kcal/mol in lines 209 and 212 (Not, Kacl/mol). Please check carefully

4. In the manuscript, the authors evaluate physicochemical parameters (such as drug-likeness properties and ADMET parameters) for their compounds, which is assumed to be the correct approach to exploring their druggability and medicinal chemistry aspects. However, the authors need to specify the significance of these parameters beforehand and report the acceptable values (or limits) of such parameters so that a clear inference can be obtained for studied compounds.

Author Response

Rebuttal Letter

Dear Editor,     

On behalf of the authors, we are delighted for considering our work for publishing in Molecules after revision and much thankful for the editor’s and reviewers’ diligent effort in helping to improve our manuscript. In the following document we clarify the inquiries raised during the reviewing cycle and provide our responses for each one raised. The reviewers’ comments were taken into consideration and a detailed clarification is addressed to each reviewers note herein and the corrections were highlighted.

No.

Reviewer 2

Responses

1

The presented study describes the synthesis of sulfamerazine derivatives as 3CLpro,  cPLA2, and sPLA2 inhibitors, and synthesis, biological testing and in silico experiments were performed.

1. The choice of Baicalein (as control) for in silico studies needs more rationality as the structure of designed compounds (6, as representative compound) does not match structurally with the structure of Baicalein (or have distinctive scaffolds); therefore, direct comparison with Baicalein requires more structural information.

For example, Baicalein is a flavonoid, while compound 6 has a hybrid structure and a distinctive structure than Baicalein. Therefore, comparing their binding pose and binding energies with each other seems trivial.

Great thanks for your remark. Please be notified that Baicalein and the tested compounds as quinazoline-based derivatives have some structural moieties similarities. “4H-chromen-4-one” in Baicalein represents pharmacophoric equivalent to “quinazoline” in the tested compounds through “ring variation”

Additionally, we compare the binding interactions with the key amino acids as the effective and fair tool for interpreting docking results between the tested compounds and the standard.

2

Comment specific to the author's statement, "All compounds exhibited interaction biding mode, they formed H-bond with Gly 143, some compounds exhibited additional interactions of induced-induced interactions with His 41."

(I) Please correct the typographical error "BINDING."

(ii) "H-bond with Gly 143" needs more information (was it the backbone of Gly 143 to which compounds are interacting)?

(iii) "induced-induced interactions" what are induced-induced interactions? Are authors talking here about pi-pi interaction, it seems quite confusing. Please specify such interaction influence on the molecular binding conformation of a molecule.

-        Word binding was corrected all throughout the manuscript.

-        H-bond type was clarified to be (H-bond acceptor) with the backbone. As Glycine doesn’t have a side chain.

-        induced-induced interactions were clarified to be Pi-Pi (π–π) interaction

3

 the author made typographical errors in various places, such as it must be Kcal/mol in lines 209 and 212 (Not, Kacl/mol). Please check carefully

Done. Binding energy unit Kcal/mol was adjusted throughout the manuscript

4

 In the manuscript, the authors evaluate physicochemical parameters (such as drug-likeness properties and ADMET parameters) for their compounds, which is assumed to be the correct approach to exploring their druggability and medicinal chemistry aspects. However, the authors need to specify the significance of these parameters beforehand and report the acceptable values (or limits) of such parameters so that a clear inference can be obtained for studied compounds.

Done. we added acceptable values (or limits) of such parameters and the application of these tested parameters for the oral bio availability of tested compounds.

-        Also, we added 5 effective references 29-33 (green colour), and all references from 29 was shifted

Round 2

Reviewer 1 Report (Previous Reviewer 1)

The authors have satisfactorily addressed most of my concerns, and I have no further suggestions. I believe the paper is acceptable for publication in the Molecules.

This manuscript is a resubmission of an earlier submission. The following is a list of the peer review reports and author responses from that submission.

Round 1

Reviewer 1 Report

In the current work, authors synthesized several novel anti-inflammatory quinazolines having a sulfamerazine moiety as a new 3CLpro, cPLA2, and sPLA2 inhibitors. Although authors responded well to the maint points for revision, I should recommend "reject" due to the low activity level of compounds. Because the IC50 value of the target compounds 4 6 and 12 against SARS CoV 2 23 main protease were found as 35.43, 30.70, 27.96 and 38.46 μM, respectively, which is not promising as proposed. The topic is original and address a specific gap in the field. A standard must be chosen according to its IC50 value for the targets, authors should investigate it I am sure they can find a better standard than ivermectin. The figures still look blurry maybe they can use some other visualization programs to increase the quality. The references are appropriate. The conclusion is far beyond the goals and sentences “may, might” could be used for these results of compounds.

Minor editing of English language required.

Author Response

Rebuttal Letter

Dear Editor,     

On behalf of the authors, we are delighted for considering our work for publishing in Molecules after revision and much thankful for the editor’s and reviewers’ diligent effort in helping to improve our manuscript. In the following document we clarify the inquiries raised during the reviewing cycle and provide our responses for each one raised. The reviewers’ comments were taken into consideration and a detailed clarification is addressed to each reviewers note herein and the corrections were highlighted.

No.

Reviewer 1

Responses

1

 In the current work, authors synthesized several novel anti-inflammatory quinazolines having a sulfamerazine moiety as a new 3CLpro, cPLA2, and sPLA2 inhibitors. Although authors responded well to the maint points for revision, I should recommend "reject" due to the low activity level of compounds. Because the IC50 value of the target compounds 4 6 and 12 against SARS CoV 2 23 main protease were found as 35.43, 30.70, 27.96 and 38.46 μM, respectively, which is not promising as proposed.

-        Last years many reports  published a new compound with promising inhibitors against SARS CoV-2 main protease with IC50s 0.2 μM to 23 μM and 40 uM

Example: 1- Coelho C, Gallo G, Campos CB, Hardy L, Würtele M (2020) Biochemical screening for SARS-CoV-2 main protease inhibitors. PLoS ONE 15(10): e0240079. https://doi.org/10.1371/journal.pone.0240079

2- Gupta S, Singh AK, Kushwaha PP, Prajapati KS, Shuaib M, Senapati S, Kumar S. Identification of potential natural inhibitors of SARS-CoV2 main protease by molecular docking and simulation studies. J Biomol Struct Dyn. 2021 Aug;39(12):4334-4345. doi: 10.1080/07391102.2020.1776157.

3. Ebselen, disulfiram, carmofur, PX-12, tideglusib, and shikonin are non-specific promiscuous SARS-CoV-2 main protease inhibitors. Chunlong Ma, Yanmei Hu, Julia Alma Townsend, Panagiotis I. Lagarias, Michael Thomas Marty, Antonios Kolocouris, Jun Wang. bioRxiv 2020.09.15.299164; doi: https://doi.org/10.1101/2020.09.15.299164

Now published in ACS Pharmacology & Translational Science doi: 10.1021/acsptsci.0c00130

 Our future plan will be aimed to prepare our compounds 4, 6, and 12 in nanoform to elevate their antiviral activity

2

The topic is original and address a specific gap in the field. A standard must be chosen according to its IC50 value for the targets, authors should investigate it I am sure they can find a better standard than ivermectin.

We selected Ivermectin and we made comparison between it and our new prepared compounds because Ivermectin is an FDA-approved broad-spectrum antiparasitic agent with demonstrated antiviral activity against a number of DNA and RNA viruses, including severe acute respiratory syndrome coronavirus 2 (SARS-CoV-2) and approved in several papers.

Example:

1.     Formiga FR, Leblanc R, de Souza Rebouças J, Farias LP, de Oliveira RN, Pena L. Ivermectin: an award-winning drug with expected antiviral activity against COVID-19. J Control Release. 2021 Jan 10;329:758-761. doi: 10.1016/j.jconrel.2020.10.009.

2.     Heidary F, Gharebaghi R. Ivermectin: a systematic review from antiviral effects to COVID-19 complementary regimen. J Antibiot (Tokyo). 2020 Sep;73(9):593-602. doi: 10.1038/s41429-020-0336-z.

3.     Zaidi, A.K., Dehgani-Mobaraki, P. The mechanisms of action of ivermectin against SARS-CoV-2—an extensive review. J Antibiot 75, 60–71 (2022). https://doi.org/10.1038/s41429-021-00491-6

Also, the inhibitory activity of compounds 4,6, 12 against SARS CoV 2 were more pronounced than ivermectin.

3

The figures still look blurry maybe they can use some other visualization programs to increase the quality. The references are appropriate.

We did our best to improve the quality of the figures

4

The conclusion is far beyond the goals and sentences “may, might” could be used for these results of compounds.

Done, and the conclusion was rewritten

Reviewer 2 Report

The authors of the manuscript entitled "Structure activity relationship and molecular docking of some quinazolines bearing sulfamerazine moiety as new 3CLpro, cPLA2, sPLA2 inhibitors" described the difficult and multi-step synthesis and inhibition studies on 3CLpro, 2 cPLA2, sPLA2 of a new quinazoline derivative. The manuscript contains interesting information and research results, but it should be reorganised. Synthesis descriptions and spectroscopic results, including elemental analysis, should be moved to the Experimental section. Elemental analysis, yields and melting points are unnecessarily in the table, these are not extraordinary results to be displayed in the table, just basic characteristics of the chemical substance.

Authors should review and follow the guidance on the description of synthesis and characterization of new substances.

Furthermore, the authors should also revise the results of NMR analysis and descriptions of MS spectra (for substance 3, the peak with 100% intensity is 287 and not 387). The 1H NMR spectra are of very poor quality and, in my opinion, not well interpreted. For example, for substance 3, the authors attribute the peak with a shift of 2.23 to the CH3 group (the peak has an integration of 42.7) and to the peak with a shift of 3.7 to the OCH3 group (the peak has an integration of 3.7) - the same number of protons and such a different integration. Some signals in the 1H NMR spectra were not assigned to rare protons at all. For substance 12, it is difficult to find the appropriate peaks at all and to determine on what basis the authors assigned them. On the basis of the analyses included in the supplement, it can be said that the tested substances are not pure.

The calculated content of elements should be added to the results of the elemental analysis.

You should also correct the units in Figure 2 - Figure 4 - the graphs are uM instead of µM. In addition, standard deviations should be added for the values in Table 2.

Author Response

Rebuttal Letter

Dear Editor,     

On behalf of the authors, we are delighted for considering our work for publishing in Molecules after revision and much thankful for the editor’s and reviewers’ diligent effort in helping to improve our manuscript. In the following document we clarify the inquiries raised during the reviewing cycle and provide our responses for each one raised. The reviewers’ comments were taken into consideration and a detailed clarification is addressed to each reviewers note herein and the corrections were highlighted.

No.

Reviewer 2

Responses

1

The authors of the manuscript entitled "Structure activity relationship and molecular docking of some quinazolines bearing sulfamerazine moiety as new 3CLpro, cPLA2, sPLA2 inhibitors" described the difficult and multi-step synthesis and inhibition studies on 3CLpro, 2 cPLA2, sPLA2 of a new quinazoline derivative. The manuscript contains interesting information and research results, but it should be reorganised. Synthesis descriptions and spectroscopic results, including elemental analysis, should be moved to the Experimental section.

Done, the synthesis description was moved to the Experimental section.

2

Elemental analysis, yields and melting points are unnecessarily in the table, these are not extraordinary results to be displayed in the table, just basic characteristics of the chemical substance.

Done, Elemental analysis, yields and melting points were moved to the Experimental section and table (1) was moved paper file to supplementary file.

-        Tables order was reorganized

3

Authors should review and follow the guidance on the description of synthesis and characterization of new substances.

Done

4

Furthermore, the authors should also revise the results of NMR analysis and descriptions of MS spectra (for substance 3, the peak with 100% intensity is 287 and not 387). The 1H NMR spectra are of very poor quality and, in my opinion, not well interpreted. For example, for substance 3, the authors attribute the peak with a shift of 2.23 to the CH3 group (the peak has an integration of 42.7) and to the peak with a shift of 3.7 to the OCH3 group (the peak has an integration of 3.7) - the same number of protons and such a different integration. Some signals in the 1H NMR spectra were not assigned to rare protons at all. For substance 12, it is difficult to find the appropriate peaks at all and to determine on what basis the authors assigned them. On the basis of the analyses included in the supplement, it can be said that the tested substances are not pure.

Done

-        According to the 1H NMR spectra of substance 3,  the difference  in integration is non-significant and the formation of compound 3, also, the structure of compound 3 were supported by elemental analysis, IR and Mass spectral data.

-        Also, elemental analysis, IR and Mass spectral clear data of compound 12 is sharp and approved its formation.

-        In addition, the preparation of our compounds is so easy and Schematic mechanism of synthesis the target compounds in figures 16a&b is logic.

5

The calculated content of elements should be added to the results of the elemental analysis.

Done , calculated content of elements was moved to the results section

6

You should also correct the units in Figure 2 - Figure 4 - the graphs are uM instead of µM.

Done

7

In addition, standard deviations should be added for the values in Table 2.

Done , table 2  was changed to table 1 after the  movement of table 1 to  the supplementary file (according to comment No. 2) and standard deviations were added

Round 2

Reviewer 1 Report

In the current work, authors synthesized several novel anti-inflammatory quinazolines having a sulfamerazine moiety as a new 3CLpro, cPLA2, and sPLA2 inhibitors. Based on revision, my decision did not change. The IC50 value of the target compounds 4 6 and 12 against SARS CoV 2 23 main protease were found as 35.43, 30.70, 27.96 and 38.46 μM, respectively, which is not promising as proposed. At this point, they proposed some studies but they are not evidence to their low activity. They said they did best for improving the quality of figures but these figures are even worse than previous figures. They also did not rewrite the conclusion. Authors did not make the revision that I asked for. Therefore, I recommend “reject”.

Moderate editing of English language required

Reviewer 2 Report

The version of the manuscript submitted for re-evaluation is no different from the previous one.